# Lights and Shadows of Microbiota Modulation and Cardiovascular Risk in HIV Patients

**DOI:** 10.3390/ijerph18136837

**Published:** 2021-06-25

**Authors:** Pablo Villoslada-Blanco, Patricia Pérez-Matute, José A. Oteo

**Affiliations:** 1Infectious Diseases, Microbiota and Metabolism Unit, Infectious Diseases Department, Center for Biomedical Research of La Rioja (CIBIR), 26006 Logroño, La Rioja, Spain; pvilloslada@riojasalud.es; 2Unidad Predepartamental de Enfermería, Universidad de La Rioja (UR), 26006 Logroño, La Rioja, Spain; 3Infectious Diseases Department, Hospital Universidad San Pedro-Center for Biomedical Research of La Rioja (CIBIR), 26006 Logroño, La Rioja, Spain; jaoteo@riojasalud.es

**Keywords:** HIV infection, cardiovascular risk, gut microbiota, prebiotics, probiotics, symbiotics, fecal transplantation

## Abstract

Human immunodeficiency virus (HIV) infection is associated with premature aging and the development of aging-related comorbidities, such as cardiovascular disease (CVD). Gut microbiota (GM) disturbance is involved in these comorbidities and there is currently interest in strategies focused on modulating GM composition and/or functionality. Scientific evidence based on well-designed clinical trials is needed to support the use of prebiotics, probiotics, symbiotics, and fecal transplantation (FT) to modify the GM and reduce the incidence of CVD in HIV-infected patients. We reviewed the data obtained from three clinical trials focused on prebiotics, 25 trials using probiotics, six using symbiotics, and four using FT. None of the trials investigated whether these compounds could reduce CVD in HIV patients. The huge variability observed in the type of compound as well as the dose and duration of administration makes it difficult to adopt general recommendations and raise serious questions about their application in clinical practice.

## 1. Introduction

In the last few decades, the life expectancy of patients with human immunodeficiency virus (HIV) infection has significantly increased due to improvements in clinical management and, specially, the extended use of antiretroviral therapy (ART). HIV infection is considered a chronic disease and life expectancy is shorter that observed in non-infected people and the general population [1]. This shorter life expectancy is associated with premature and accelerated aging [2] as well as the continuous activation of the immune system and low-grade chronic inflammation [3]. Thus, HIV patients have higher rates of cardiovascular disease (CVD), mainly atherosclerosis, and non-acquired immune deficiency syndrome (AIDS)-defining cancers, frailty, kidney or liver diseases, and neurological complications, such as dementia, compared with uninfected individuals of the same age [4,5,6,7,8]. Recent studies have looked beyond traditional cardiovascular risk (CVR) factors and have expanded the view towards immunology, intestinal microbiome, red blood cells, and endothelial function disorders [9,10,11,12]. Thus, the presence of factors other than “classical” CVR factors requires in-depth investigation of the physiopathology of these processes in the context of HIV infection. In addition, interventions to modify both traditional and “novel” risk factors are needed in people with HIV to reduce future adverse events, counteract the consequences of accelerated aging, and improve quality of life [13].

In this context, the gut microbiota (GM) is a “novel” factor recently associated with cardiovascular events in HIV-infected patients as well as the general population. The term microbiota describes the community of microorganisms that coexists in a specific environment, including bacteria, archaea, viruses, and some unicellular eukaryotes [14]. Several studies have demonstrated that alterations in the GM play an important role in health and disease [15]. The development of next-generation sequencing technologies has allowed the identification of new markers of the physiological and pathological states of each individual, enabling an approximation and description of the GM in different medical disorders [16,17]. However, it is still unclear whether GM modifications are the cause or consequence of alterations in health and each disorder should be examined in depth. HIV infection is characterized by a set of structural changes of the gut epithelial barrier, immunological shifts, and modifications to the composition and functionality of the GM. Under normal physiological conditions, microorganisms are found in the intestinal lumen interacting with the intestinal cells in a state of symbiosis; however, when HIV infection occurs, depletion of CD4^+^ T lymphocytes occurs in the gut-associated lymphoid tissue. This induces rupturing of the epithelial barrier, which triggers alterations in the intestinal lumen and the composition of the microbiota (at least at the bacterial level) [18]. This dysbiosis favors the passage of microorganisms and their components to the lamina propria and, hence, to the circulation, which is known as bacterial translocation (BT), leading to subsequent intestinal and systemic inflammation [19] (Figure 1). This immune activation and persistent chronic inflammation are observed in HIV-infected patients and are strongly predictive of atherosclerosis [20] and associated with the development of non-AIDS events, which increase morbidity and mortality and substantially reduce patients’ quality of life [19,21]. ART is not able to completely reverse this situation. Preliminary studies using small numbers of patients have shown an improvement in GM composition after treatment with integrase inhibitors [18].

There is interest currently in strategies focused on reversing or modulating GM composition and/or functionality along with standard care (i.e., ART). The use of prebiotics, probiotics, symbiotics, and fecal transplantation (FT) have emerged as “novel” promising therapies to control or reverse the changes observed in GM composition in HIV infection and reduce the incidence of comorbidities, such as cardiovascular events or obesity [23]. Nevertheless, despite the good reputation of these interventions in several recurrent diseases other than HIV infection, concerns have emerged regarding the use of GM modulators in clinical practice. Thus, solid scientific evidence is required to demonstrate their use in specific populations, such as HIV-infected people. It is also essential to establish the optimal dose, route, administration frequency, and exact composition of the therapies.

The objective of the present review was to investigate whether there is strong scientific evidence (based on well-designed clinical trials) to support the use of prebiotics, probiotics, symbiotics, and FT to modify GM in HIV-infected patients.

## 2. Search Strategies, Inclusion/Exclusion Criteria, and Quality Assessment

A search was performed using PubMed to identify all clinical trials (using the Clinical Trial filter) including adult humans (age ≥ 19 years) that were published within the last 10 years. We limited the search to clinical trials since they are at the top of the “pyramid of scientific evidence,” just below meta-analysis and systematic reviews [24]. Meta-analyses and systematic reviews were also used to confirm the clinical trials chosen for this review. A 10-year period was chosen because 657 of the 688 results obtained from PubMed using the keywords “microbiota HIV” were published in the last 10 years, which means a large and representative proportion (95.49%) of the total investigation developed in this field. Furthermore, a search in ClinicalTrials.gov (last access: 1 March 2021) was also carried out and these studies were included and described in the present review. Following these criteria, we identified three studies concerning prebiotics, 25 concerning probiotics, six concerning symbiotics, and four concerning FT. All clinical trials analyzed passed the Critical Appraisal Skills Programme.

Eight published studies were identified using PubMed concerning the effects of prebiotics in HIV-infected patients within the last 10 years. Only five were derived from clinical trials, among which three were excluded because they were related to symbiotics and were therefore included in the symbiotics section of this review. Among the two remaining articles, only one (NCT01838915) was registered with ClinicalTrials.gov. In addition, another clinical trial (NCT04058392) appeared in this website. Table 1 shows these two clinical trials registered in ClinicalTrials.gov; however, we have discussed the results obtained from all three clinical trials that focused on prebiotics and HIV infection independently of registration on the website.

PubMed identified 35 published studies concerning probiotics in HIV-infected patients in the last 10 years. Only 15 were derived from clinical trials, among which one was excluded because it only contained data relating to symbiotics and, therefore, was incorporated in the symbiotics section of the review. Another three publications were dismissed because they did not include HIV-infected patients. We included two more clinical trials that did not appear in the PubMed search but met the inclusion criteria [25,26]. We appended one more clinical trial from the unique meta-analysis and/or systematic review published [27] that met the selection criteria and was not included previously [28]. Among the 14 remaining articles, only five were registered in ClinicalTrials.gov and a further 11 clinical trials appeared in this website. Table 2 shows all the clinical trials registered with ClinicalTrials.gov, and all 25 clinical trials that focused on probiotics and HIV infection are included in this section.

Finally, five studies concerning the effects of symbiotics (a combination of synergistically acting prebiotics and probiotics [29]) in HIV-infected patients in the last 10 years were identified using PubMed, of which only four were derived from clinical trials. Among these four studies, three were registered with ClinicalTrials.gov. Two more clinical trials were included in this website. Table 3 shows the clinical trials concerning symbiotics registered with ClinicalTrials.gov. A discussion of all six clinical trials has been included in this review. Figure 2 shows a flow chart of the study selection process for this review.

## 3. Microbiota Modulation to Reduce Cardiovascular Risk in HIV Patients: Scientific Evidence from Clinical Trials

### 3.1. Prebiotics

Administration of prebiotics in HIV patients can modify the composition of the microbiota at several taxonomical levels. However, this influence is only observed in naive patients [30], possibly because the time since diagnosis of HIV infection is shorter in naive patients compared with ART-treated patients. Thus, it is possible that virus-induced changes at the intestinal level are mild in naive patients and easier to reverse, whereas ART-treated patients could present a more established and stabilized GM. Specifically, naive HIV patients treated for six or 12 weeks with different combinations of prebiotics [short-chain galacto-oligosaccharides/long-chain fructo-oligosaccharides/pectin hydrolysate-derived acidic oligosaccharides/glutamine (scGOS/lcFOS/pAOS/glutamine)) showed significant increases in *Bifidobacterium* genus (2.8–15.7%, *p* = 0.0007) [37] and the phyla Firmicutes (including *Faecalibacterium*, *Catenibacterium*, *Blautia* and *Eubacterium*) and Actinobacteria (including *Collinsella* and *Corynebacterium)* [30]. In contrast, statistically significant decreases in *Clostridium lituseburense*/*Clostridium histolyctum* (including *Clostridium perfringens* and *Clostridium difficile*) (0.016–0.002%, *p* = 0.009) [37] and *Eubacterium rectale*/*Clostridium coccoides* (*p* = 0.035) have also been demonstrated in these patients [30]. Table 4 shows the bacteria whose relative abundance was modified by these interventions and some of their functions in health and disease.

Administration of scGOS/lcFOS/pAOS for 12 weeks was associated with a decrease in BT markers, such as sCD14 and lipopolysaccharide (LPS) in naïve patients [37]. However, no statistically significant differences were observed in these BT markers using a similar mixture of prebiotics (scGOS/lcFOS/glutamine) for six weeks [30]. These discrepancies could be due to different durations of the interventions (12 weeks vs. 6 weeks). Thus, the microbiota is stable in adulthood, and longer periods of administration are required to modify it.

Interestingly, while no changes were observed in the viral load or number of CD4^+^ T lymphocytes after prebiotic administration, a decrease in the activation of these lymphocytes was observed after 12-week administration of scGOS/lcFOS/pAOS (0.52%–0.27% in naive patients, *p* < 0.01) [37] or after a six-week intervention with scGOS/lcFOS/glutamine [30]. This suggests that, although the intervention was unable to restore the number of CD4^+^ T lymphocytes, it facilitated the reduction of the immune system activation, which could be beneficial for these patients, given the persistent chronic inflammation characteristics of HIV infection.

In these two studies, administration of prebiotics was safe and there were no observed modifications to hepatic or renal function [30,37]. Nevertheless, in terms of tolerability, one study reported complications (mainly diarrhea), which disappeared four weeks after the start of the intervention [37]. Administration of prebiotics in HIV patients was able to modify the composition of the microbiota and reduce the activation of CD4^+^ T lymphocytes, while the effects on markers of inflammation and BT remain unclear. No studies have investigated whether these changes are able to reduce the incidence of comorbidities in this population and, specifically, whether they are able to reduce CVR. Moreover, the type of compound and the duration of administration is a variable to consider when drawing conclusions and making general recommendations. Further clinical trials are required to evaluate the effects of these compounds on CVR factors in HIV-infected patients as well as follow-up studies to corroborate a significant reduction in cardiovascular events in patients who showed improvements in inflammation and intestinal dysbiosis.

### 3.2. Probiotics

The use of probiotics, such as *Lactobacillus*, *Bifidobacterium*, and *Enterococcus*, that can benefit the intestinal and immune system, may be inexpensive and clinically important as coadjutants of ART in order to reduce HIV-related morbidity and mortality [57]. The mechanisms by which probiotics may interfere with HIV were previously summarized by D’Angelo et al. [58]. In essence, probiotics can rebalance the microbiota by competing with pathogens, improving the intestinal barrier, and reducing BT, consequently decreasing systemic inflammation and restoring the mucosal immune function, lowering local inflammation [58].

Administration of probiotics in HIV-infected patients can modify the composition of the microbiota in naive and ART-treated patients. Specifically, intervention of ART-treated patients with different combinations of probiotics was associated with an increase in the relative abundance of Actinobacteria and Firmicutes phyla [250 mL/day of fermented skimmed milk supplemented with *Lactobacillus rhamnosus* GG (10^8^ CFU/mL), *Bifidobacterium animalis* subsp. *lactis* B-12 (10^8^ CFU/mL), and *Lactobacillus acidophilus* La-5 (10^7^ CFU/mL) for eight weeks] [32] as well as an increase in *Bifidobacterium* spp. (Visbiome; twice daily for 24 weeks) [33]. A statistically significant increase in the abundance of *Megamonas* and Desulfovibrionales levels after 12-week administration of capsules containing 56.5 mg of living *Saccharomyces boulardii* were also observed [59]. In contrast, a statistically significant reduction of Bacteroidetes phylum and some species of Clostridiales and *Catenibacterium* were also demonstrated after the administration of 250 mL/day of fermented skimmed milk supplemented with *L. rhamnosus* GG (10^8^ CFU/mL), *B. animalis* subsp. *lactis* B-12 (10^8^ CFU/mL) and *L. acidophilus* La-5 (10^7^ CFU/mL) for eight weeks [32], and after administration of capsules containing 56.5 mg of living *S. boulardii*, respectively [59]. In a study carried out in naive patients, a decrease in total bacterial load in stools compared with the placebo group and an increased *Bifidobacterium* spp. concentration along with a decrease in *Clostridium* spp. concentration was demonstrated after daily administration of *L. rhamnosus* HN001 plus *Bifidobacterium lactis* Bi-07 [10^9^ CFU/mL) for 16 weeks [60]. Finally, in a study with both naive and ART-treated patients, a reduction of the orders Enterobacteriales (*p* = 0.018) and Erysipelotrichales (*p* = 0.037) entirely driven by a reduction of the Enterobacteriaceae and Erysipelotrichaceae families were observed, along with an increase in an uncultured genus from the Lachnospiraceae family (*p* = 0.015) and *Ruminiclostridium* (*p* = 0.023) from the Ruminicoccaceae family following administration of two capsules daily of *L. rhamnosus* GG (10^10^ CFU) for eight weeks [61]. Table 4 shows the bacteria (with different taxonomic levels) whose relative abundance was modified in response to interventions, and some of their functions in health and disease.

Concerning inflammation, twice daily administration of a 1-g packet (packet A) containing *Streptococcus salivarius* ssp. *thermophilus* (≥2.04 × 10^14^ CFU), Bifidobacteria represented by *Bifidobacterium breve*, *Bifidobacterium*
*infantis* and *Bifidobacterium*
*longum* (≥9.3 × 10^13^), *L. acidophilus* (≥2 billion CFU), *Lactobacillus plantarum* (at least 2.20 × 10^8^ CFU), *Lactobacillus casei* (≥2.20 × 10^8^ CFU), *Lactobacillus delbrueckii* ssp. *bulgaricus* (≥3 × 10^8^ CFU) and *Streptococcus faecium* (≥3 × 10^7^ CFU) for 48 weeks was not able to reduce D-dimer levels [31]. However, administration of 250 mL/day of fermented skimmed milk supplemented with *L. rhamnosus* GG (10^8^ CFU/mL), *B. animalis* subsp. *lactis* B-12 (10^8^ CFU/mL), and *L. acidophilus* La-5 (10^7^ CFU/mL) for eight weeks led to reductions in levels of both D-dimer (*p* = 0.03) and interleukin 6 (IL-6; *p* = 0.06) compared with the placebo group [32]. Furthermore, 12-week administration of *S. boulardii* led to reduced levels of IL-6 compared with the placebo group [25]. These discrepancies could be due to the different combinations used; therefore, the positive effects of probiotics on inflammation or at least on these parameters measured [D-dimer and/or IL-6) remain unclear. In relation to the effects on BT, different combinations of probiotics (capsules containing 56.5 mg of living *S. boulardii*, packet A, and *S. boulardii*) led to decreased levels of lipopolysaccharide-binding protein (LBP) compared with that of the placebo group [25,31,59]. However, no effects were observed for sCD14 or sCD163 levels [31,32,59,61] despite the fact that sCD14 has been suggested to be a more relevant biomarker of disease progression as it reflects the host response to products of BT [62]. These results make it difficult to confirm the beneficial effects of these probiotics on BT.

On the other hand, an increase in the number of CD4^+^ T lymphocytes was observed in ART-treated patients after the administration of *L. rhamnosus* GR-1 and *Lactobacillus*
*reuteri* RC-14 for 25 weeks [63]. Moreover, an increased percentage of CD4^+^ T lymphocytes was also observed in the probiotic group after administration of packet A [31]. Administration of *Bacillus coagulans* GBI-30, 6086 (Ganeden BC^30^; 2 × 10^12^ CFU) once daily for 90 days was also associated with a significant increase in the percentage of CD4^+^ T lymphocytes compared with the placebo group (+2.8% vs. −1.8%, *p* = 0.018), although the concentrations were generally unchanged in each group [28]. Nonetheless, no increase in CD4^+^ T lymphocytes was observed after administration of 125 mL of yoghurt fortified with *L. rhamnosous* GR-1 (1.23 × 10^9^ CFU/mL) once daily for four weeks, possibly due to the shorter duration of the intervention [64]. Interestingly, a decrease was observed in the activation markers, CD38 and HLA-DR, after the administration of packet A [31] and Visbiome for 24 weeks [65].

In all these studies, the administration of probiotics was safe and showed no side effects or modifications in hepatic or renal functions [26,28,31,63,64,65,66,67].

In summary, there is some scientific evidence showing that administration of probiotics in HIV patients can modify the composition of the microbiota, increase the number of CD4 T^+^ lymphocytes and decrease their activation, and reduce LBP levels. Although there are more studies into probiotics than prebiotics, these studies do not address the potential effects of probiotics on CVR in patients with HIV. Moreover, it is necessary to assess whether the changes observed in markers of inflammation and BT after the ingestion of probiotics could lead to improvements in the incidence of comorbidities and, specifically, CVR, in this population. In addition, the huge variability in the type of compound used, dose, and duration administration makes it impossible to make verified and confirmed conclusions. Therefore, studies using large cohorts are required to obtain a good description of the bacterial species that are decreased in HIV-infected patients. These depleted species would be good candidates for ingestion as probiotics to restore GM, but they should pass the requirements imposed by the Food and Agriculture Organization of the United Nations and the World Health Organization to ensure that they are safe to use. Finally, the effects of these probiotics on markers of inflammation, BT, and CVR should be assessed in HIV-infected patients.

### 3.3. Symbiotics

The administration of symbiotics in HIV patients has been shown to modify GM composition. Specifically, 16-week administration of 10 g of agavins from *Agave tequilana* plus *L. rhamnosus* HN001 and *B. lactis* Bi-07 (10^9^ CFU/mL) in naive patients led to a statistically significant decrease in total load of bacteria in stools compared with the placebo group [60]. Similarly, in another study carried out in ART-treated patients, increased levels of *L. plantarum* and *Pediococcus pentosaceus* were detected after four-week administration of Synbiotic 2000 compared with the placebo group [35]. However, administration of PMT25431 for 48 weeks in naive patients who initiated ART did not lead to significant changes in GM [34]. The functions of these bacteria in health and disease are summarized in Table 4.

In terms of inflammation, a study of naive patients demonstrated a statistically significant decrease in IL-6 levels compared with at the beginning of the intervention following administration of 10 g of agavins from *A. tequilana* plus *L. rhamnosus* HN001 and *B. lactis* Bi-07 (10^9^ CFU/mL) [60]. However, administration of PMT25431 over 48 weeks in naive patients who initiated ART did not induce additional changes in IL-6 levels [34]. Moreover, there were no effects on plasma levels of TNF-α, IL-1β, IL-10, sCD14, or sCD163 in naive HIV-infected patients after the administration of 10 g of agavins from *A. tequilana* plus *L. rhamnosus* HN001 and *B. lactis* Bi-07 (10^9^ CFU/mL) [60]. Administration of PMT25341 over 48 weeks showed no effects in naive patients who initiated ART [34], and no effects were observed in ART-treated patients after the administration of Synbiotic 2000 over four weeks [35].

Increased levels of CD4^+^ T lymphocytes were observed in naive HIV-infected patients after the administration of 10 g of agavins compared with levels at the beginning of treatment [60]. However, no effects were observed after four-week administration of Synbiotic 2000 in ART-treated patients [35]. These discrepancies could be due to different lengths of intervention, different products used, and the fact that, in the second study, patients were treated with ART and their GM may have been more established, as previously mentioned. Finally, administration of PMT25341 for 48 weeks in naive HIV-infected patients who initiated ART did not induce additional changes compared with those exerted by ART in terms of levels and activation of CD4^+^ T lymphocytes [34].

In all these studies, administration of symbiotics was safe and without side effects [35,36,60]. Furthermore, administration of LACTOFOS twice daily for 30 weeks in naive patients significantly reduced the incidence of diarrhea (−21.8%), nausea and/or vomiting (−28.8%), constipation (−13.2%), and dyspepsia (−24.5%) [36].

In general terms, these results showed that administration of symbiotics in HIV-infected patients modified the composition of the microbiota, while the effects on the number of CD4^+^ T lymphocytes and the markers of inflammation and BT were not as clear as in the case of probiotics, which is interesting considering that symbiotics are a mixture of probiotics and prebiotics. However, these discrepancies could be due to the use of different probiotics and/or different durations of administration. Similarly, as observed with prebiotics and probiotics, no studies have focused on the long-term consequences of these changes. As there is a huge variability in the type of compound, dose, and duration of administration, it is not possible to draw conclusions and make general recommendations.

### 3.4. Fecal Transplantation

FT is the transfer of stools from a healthy donor into a patient with a disease characterized by a significantly altered microbiome. Its objective is to restore the microbiota and, therefore, treat the disease [68]. FT has gained acceptance as a safe and effective treatment for recurrent *Clostridioides difficile* infection (CDI)-associated diarrhea [69,70,71,72,73]. While HIV patients treated for CDI using FT did not show any adverse events, 14% of patients with inflammatory bowel disease experienced a disease flare that required hospitalization in some cases [74]. Therefore, FT could involve an uncontrollable risk, particularly in immunocompromised patients, if donor samples are not checked [75,76,77]. Thus, more well-designed studies and clinical trials are required.

To date, four clinical trials have been registered with ClinicalTrials.gov (Table 5) and the results of one trial have been published and was identified in PubMed (NCT03008941). In this clinical trial, ART-treated patients were orally administered 10 capsules of fecal content for one week followed by six-week treatment with five capsules. A significant increase in alpha diversity was observed in these patients compared with those receiving placebo. Long-lasting effects during a 48-week follow up were also observed. Furthermore, changes in beta diversity and increased abundance of different members of the Lachnospiraceae family (*Anaritides* spp., *Blautia* spp., *Dorea* spp., and *Fusicatenibacter* spp.) and the Ruminicoccaceae family were observed in patients treated with fecal microbiota transplantation (FMT). A statistically significant decrease was also observed in intestinal fatty acid-binding protein (IFABP), which is a biomarker of intestinal injury that independently predicts mortality in treated HIV-patients [78,79]. However, levels of CD4 T^+^ lymphocytes, CD4/CD8 ratio, lymphocyte markers of immune activation, and plasma markers of inflammation and BT were not affected by FMT. Finally, no serious adverse events were detected during the intervention or follow up [80]. The involvement of these bacteria in health and disease is shown in Table 4.

Finally, Table 6 includes the list of “biotics” analyzed and their definition or composition.

## 4. Conclusions

Among all the GM modulatory strategies analysed, our review identified probiotics as the most beneficial therapy in HIV patients to improve their immune and inflammatory states. However, very few studies have focused on their effects on specific CVR markers in HIV patients or the long-term consequences of these effects on inflammation. Follow-up studies are required to determine whether the effects of probiotics on BT and inflammation lead to statistically significant reductions in cardiovascular events with improvements in inflammation, intestinal dysbiosis and CVR factors. In addition, huge variability in the type of compound, dose, and length of administration was observed among the trials, which raises serious questions about the practical value of these approaches. Long-term interventions are the most potent methods to modify the GM and reverse inflammation.

In summary, HIV-infected patients have significantly increased rates of cardiovascular events, even after controlling for traditional CVR factors and this significantly affects their quality of life and represents a significant increase in healthcare costs. The use of probiotics or other GM modulatory strategies along with the standard of care (i.e., ART) could be a good strategy to modify CVR factors in this population. However, to date, the results obtained from clinical trials are not conclusive and more prospective controlled studies with larger number of patients and with standardized dose and duration of therapies are needed for appropriate applications in clinical practice. Patient follow-up is also required to confirm the results.

## Figures and Tables

**Figure 1 ijerph-18-06837-f001:**
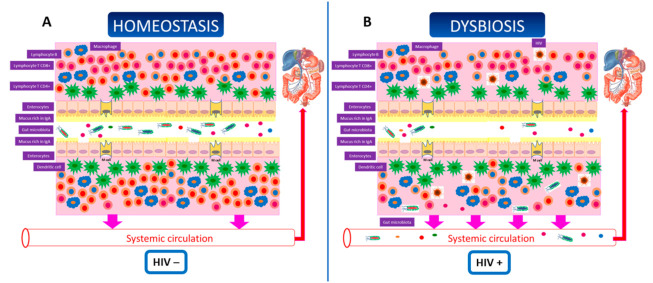
Under normal physiological conditions, microorganisms in the intestinal lumen interact with intestinal cells in a state of symbiosis; however, when HIV infection occurs, depletion of CD4^+^ T lymphocytes occurs in the gut-associated lymphoid tissue. This is accompanied by rupturing of the epithelial barrier, which triggers alterations in the intestinal lumen as well as the composition of the microbiota (at least at the bacterial level) [18]. This dysbiosis favors the passage of microorganisms and their components to the lamina propria and, therefore, to the circulation, which is known as bacterial translocation (BT), leading to subsequent intestinal and systemic inflammation [19]. (**A**) Gut homeostasis. (**B**) Gut dysbiosis after HIV infection. Modified from Pérez-Matute P et al. [22].

**Figure 2 ijerph-18-06837-f002:**
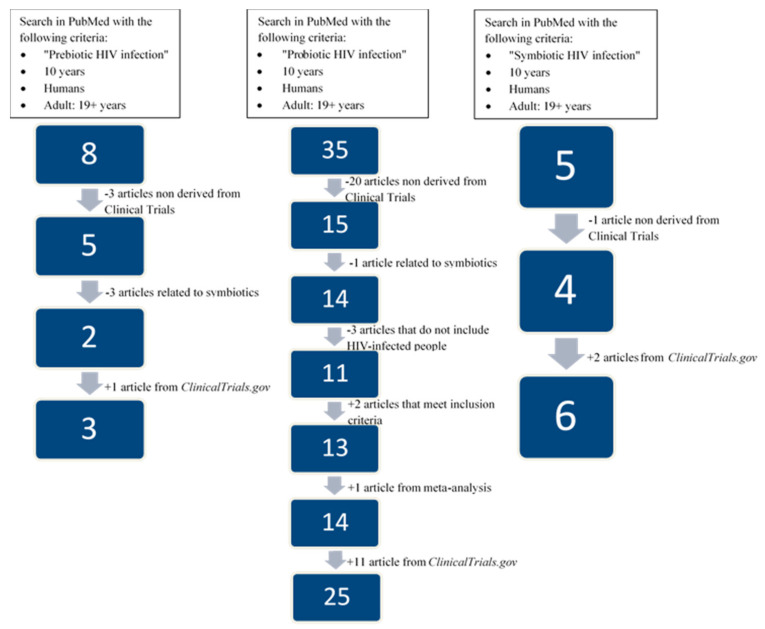
Flow chart of the study selection concerning the effects of prebiotics, probiotics, and symbiotics in HIV-infected people.

**Table 1 ijerph-18-06837-t001:** Clinical trials concerning prebiotics registered in ClinicalTrials.gov.

NCT Number	Country	Reference	Status	Intervention	Number of Participants	Age of Participants (Years)	Objective	Main Outcomes
NCT01838915	Spain	Serrano Villar et al., 2017 [30]	Completed	scGOS/lcFOS + glutamine 6 weeks	45	18–70	To assess the effects in naive and ART-treated HIV-infected patients on: safety GM composition bacterial translocation immune activation endothelial dysfunction.	Safe. Modification of GM composition in naive patients only. Decreased activation of CD4^+^ T lymphocytes in naive and ART-treated patients.
NCT04058392	Canada	–	Not yet recruiting	Camu camu (2 capsules/day) 12 weeks	22 (enrolled)	≥18	To evaluate the effects in ART-treated HIV-infected patients on: tolerance GM composition inflammation changes of gut barrier.	

ART, antiretroviral treatment; HIV, human immunodeficiency virus); GM, gut microbiota; lcFOS, long-chain fructo-oligosaccharide; scGOS, short-chain galacto-oligosaccharides.

**Table 2 ijerph-18-06837-t002:** Clinical trials concerning probiotics registered in ClinicalTrials.gov.

NCT Number	Country	Reference	Status	Intervention	Number of Participants	Age of Participants (Years)	Objective	Main Outcomes
NCT02164344	Italy	D’Etorre et al., 2015 [31]	Unknown	*S. salivarius* spp. *thermophilus* 2.04 × 10^14^ CFU, *B. breve*, *B. infantis*, *B. longum* 9.3 × 10^13^ CFU, *L. acidophilus* 2 × 10^12^ CFU, *L. plantarum* 2.2 × 10^8^ CFU, *L. casei* 2.2 × 10^8^ CFU, *L. delbrueckii* ssp. *Bulgaricus* 3 × 10^8^ CFU and *S. faecium* 3 × 10^7^ CFU 1 daily dose 48 weeks	30	18–75	To determine the effects in ART-treated HIV-infected patients on: microbial translocation-induced inflammation immune function and activation.	Safe. Decreased LBP levels. Increased CD4^+^ T lymphocytes. Decreased activation of markers CD38 and HLA-DR presented on CD4^+^ T lymphocytes.
NCT01439841	Norway and Sweden	Stiksrud et al., 2015 [32]	Completed	Biola (*L. rhamnosous* GG 10^8^ CFU, *L. acidophilus* La-5 10^7^ CFU and *Bifidobacterium animalis* subsp. *lactis* Bb-12 10^8^ CFU) 1 daily dose 8 weeks	32	≥18	To investigate the effects in ART-treated HIV-infected patients on: safety and tolerability GM composition immune activation microbial translocation disease progression.	Modification of GM composition. Decreased D-dimer and IL-6 levels.
NCT00517803	Canada	–	Completed	Probiotic yogurt (175 g) 1 daily yogurt 16 weeks	24	18–85	To compare the effects of several levels of fortified probiotic yogurt vs. placebo in HIV-infected patients with cancer on: inflammation immune function.	
NCT02448238	USA	–	Completed	VSL#3 (*S. thermophilus*, *B. breve*, *B. longum*, *B. infantis*, *L. plantarum*, *L. acidophilus*, *L. paracasei* and *L. bulgaricus* (9 × 10^11^ CFU)) 1 daily dose 12 weeks	23 (females only)	≥18	To assess the effects in naive HIV-infected Malian women on: safety, tolerability, and level of stress inflammation bacterial translocation immune function.	
NCT02764684	Denmark	–	Completed	*Lactobacillus rhamnosus* 2 daily doses 8 weeks	45	≥18	To evaluate the effects in naive HIV-infected patients on: GM composition microbial translocation lipid metabolism cardiovascular risk factors systemic and gut inflammation.	
NCT02441231	Canada	–	Unknown	Visbiome 2 daily doses (9 × 10^14^ CFU/day) 48 weeks	36 (males only)	≥19	To investigate the effects in ART-treated HIV-infected men on: microbial translocation immune cells function and activation.	
NCT00536848	Tanzania	Hummelen et al., 2010 [26]	Unknown	*Lactobacillus rhamnosus* GR-1 and *L. reuteri* RC-14 (2 × 10^9^ CFU) 1 daily dose 24 weeks	65 (females only)	18–45	To determine the effects in naive HIV-infected women with vaginosis on: diarrhea immune system bacterial vaginosis.	No improvements in bacterial vaginosis.
NCT01908049	Spain	Villar-García et al., 2015 [25]	Unknown	*S. boulardii* 3 daily doses 12 weeks	44	≥18	To assess the effects in ART-treated HIV-infected patients on: Bacterial translocation GM composition immune function viral load.	Decreased IL-6 and LBP levels.
NCT02706717	USA	–	Completed	Visbiome 1 daily dose 24 weeks	93	≥18	To evaluate the effects in ART-treated HIV-infected patients on: Inflammation	
NCT02276326	Italy	Ceccarelli et al., 2017 [33]	Recruiting	VSL#3 (*S. thermophilus*, *B. breve*, *B. longum*, *B. infantis*, *L. plantarum*, *L. acidophilus*, *L. paracasei*, and *L. bulgaricus* (9 × 10^11^ CFU)) 4 daily doses 16 weeks	20	≥18	To determine the effects in ART-treated HIV-infected patients on neurocognitive profile.	Modification of GM composition. Improved neurocognitive profile.
NCT02640625	Norway	–	Completed	*Lactobacillus rhamnosus* GG, *L. acidophilus*, *L. bulgaricus*, *B. animalis* subsp. *lactis*, and *S. thermophilus* 1 daily dose 10 weeks	60 (males only)	25–65	To assess the effects in ART-treated HIV-infected men on: safety biological effects To investigate differences between IR and INR in: microbial composition mucosal barrier function.	
NCT01184456	USA	–	Completed	GanedenBC30 (*Bacillus coagulans* GBI-30 andPTA-6086 (2 × 10^12^ CFU)) 1 daily dose 12 weeks	24	≥18	To evaluate the effects in ART-treated HIV-infected patients on bacterial translocation.	
NCT04297501	China	–	Completed	*Bifidobacterium* (3 × 10^12^) and *Lactobacillus* (10^12^) 1 daily dose 12 months	50	18–65	To investigate the effects in ART-treated HIV-infected patients on: microbial composition and diversity immune recovery and activation gut damage microbial translocation inflammation.	
NCT04297488	China	–	Not yet recruiting	*Bifidobacterium* (3 × 10^12^) and *Lactobacillus* (10^12^) 1 daily dose 6 months	20	18–65	To explore the effects in INR HIV-infected patients on: microbial composition and diversity immune recovery and activation gut damage microbial translocation inflammation.	
NCT04175223	France	–	Not yet recruiting	Vivomixx 2 daily doses 6 months	50	≥18	To evaluate the effects in ART-treated HIV-infected patients with neurocognitive disorders on: immune activation cognitive performance.	
NCT0409943	Italy	–	Recruiting	Vivomixx 4 daily doses 6 months			To assess the effects in ART-treated HIV-infected patients with HPV genital infection on: anal HPV clearance anal dysplasia.	

ART, antiretroviral treatment; CFU, colony forming unit; GM, gut microbiota; HIV, human immunodeficiency virus; HPV, human papilloma virus IL-6, interleukin-6; IR, immunological responder; INR, immunologic non-responder.

**Table 3 ijerph-18-06837-t003:** Clinical trials concerning symbiotics registered in *ClinicalTrials.gov*.

NCT Number	Country	Reference	Status	Intervention	Number of Participants	Age of Participants (Years)	Objective	Main Outcomes
NCT03009032	Spain	Serrano-Villar et al., 2019 [34]	Completed	PMT25341 1 daily dose 48 weeks	77	18–80	To investigate the effects of the intervention in naive HIV-infected patients who initiate ART on: safety immunological recovery inflammatory markers GM composition.	No effects on microbiota composition No effects on CD4^+^ T lymphocytes. No effects on markers of inflammation and bacterial translocation.
NCT00688311	USA	Schunter et al., 2012 [35]	Completed	Synbiotic 2000 (4 probiotic bacteria (10^10^ CFU/bacteria) + 4 dietary fibers (2.5 g/fiber)) 1 daily dose 4 weeks	34 (only females)	≥18	To determine the effects in ART-treated HIV-infected women on: intestinal function immune system overactivation immune function.	Safe. Modification of GM composition.
NCT03568812	Indonesia	–	Recruiting	Rillus [*L. plantarum* 8.55 mg, *S. thermophilus* 8.55 mg and *B. bifidum* 2.55 mg (10^9^ CFU) + FOS (480 mg)] 1 daily dose 12 weeks	80	18–55	To assess the effects INR HIV-infected patients on: gut mucosal integrity and immunity bacterial translocation gut inflammation.	
NCT03542786	Spain	–	Recruiting	I3.1 (*L. plantarum CECT7484, L. plantarum CECT7485* and *P. acidilactici)* + ProSheed 1 daily dose 48 weeks	40	≥18	To evaluate the effects in ART-treated HIV-infected patients on inflammaging (premature aging).	
NCT02180035	Brazil	Santos et al., 2017 [36]	Completed	*Lactobacillus paracasei*, *Lactobacillus rhamnosus*, *Lactobacillus acidophilus* and *Bifidobacterium lactis* + FOS 2 daily doses 24 weeks	290	≥19	To investigate the effects in ART-treated HIV-infected patients on lipid metabolism.	Reduced diarrhea, nausea and/or vomiting, constipation, and dyspepsia.

ART, antiretroviral treatment; CF, colony forming unit; GM, gut microbiota; HIV, human immunodeficiency virus; INR, immunological non-responder.

**Table 4 ijerph-18-06837-t004:** Bacteria whose relative abundance is modified by the interventions described in this review and their functions in health and disease.

	Phylum	Class	Order	Family	Genus	Species	Function(s)
**It INCREASE**	Firmicutes	Clostridia	Clostridiales	Ruminococcaceae	*Faecalibacterium*		One of the main butyrate producers in the gut. Plays a crucial role in gut physiology and host wellbeing [38].
					*Ruminiclostridium*		Significantly enriched in healthy donors compared with people with metabolic issues [39].
				Lachnospiraceae	*Blautia*		Contributes to alleviation of inflammatory diseases and metabolic diseases and has antibacterial activity against specific microorganisms [40].
					*Anaritides*		Unknown.
					*Dorea*		Capable of degrading polysaccharides, oligosaccharides and sugars [41].
					*Fusicatenibacter*		Butyrate production [42].
				Eubacteriaceae	*Eubacterium*		Butyrate production. Plays a critical role in energy homeostasis, colonic motility, immunomodulation, and suppression of inflammation in the gut. Bile acid and cholesterol transformations [43].
		Erysipelotrichia	Erysipelotrichales	Erysipelotrichaceae	*Catenibacterium*		Polysaccharide degradation [44].
		Negativicutes	Selenomonadales	Veillonellaceae	*Megamonas*		Production of the SCFAs acetate, propionate, and butyrate production, which are beneficial for health [45].
		Bacilli	Lactobacillales	Lactobacillaceae	*Lactiplantibacillus*	*plantarum*	Production of bacteriocins, antimicrobial peptides [46].
					*Pediococcus*	*pentosaceu*	Alleviation of intestinal inflammation by maintaining the intestinal integrity and modulating the immunological profiles, gut microbiome, and metabolite composition [47].
	Actinobacteria	Actinobacteria	Coriobacteriales	Coriobacteriaceae	*Collinsella*		Butyrate production [48].
			Actinomycetales	Corynebacteriaceae	*Corynebacterium*		Cause of significant infections [49].
			Bifidobacteriales	Bifidobacteriaceae	*Bifidobacterium*		Numerous positive health benefits [50].
	Proteobacteria	Deltaproteobacteria	Desulfovibrionales				Sulfate reduction [51].
**It DECREASE**	Firmicutes	Clostridia	Clostridiales	Clostridiaceae	*Clostridium*	*lituseburense*	Association with a range of human and animal diseases [52].
						*histolyctum*	
						*perfringens*	
						*coccoides*	
				Peptostreptococcaceae	*Clostridioides*	*difficile*	Causes diarrhea, fulminant colitis, and death [53].
				Eubacteriaceae	*Eubacterium*	*rectale*	Contributes to colorectal cancer by promoting colitis and inflammation [54].
		Erysipelotrichia	Erysipelotrichales	Erysipelotrichaceae	*Catenibacterium*		Polysaccharide degradation [44].
	Proteobacteria	Gammaproteobacteria	Enterobacteriales	Enterobacteriaceae			Causes many nosocomial infections, community-acquired infection, respiratory infections, soft tissue infections, osteomyelitis, and endocarditis [55].
	Bacteroidetes						Significant clinical pathogens [56].

SCFA, short-chain fatty acid.

**Table 5 ijerph-18-06837-t005:** Clinical trials concerning fecal transplantation registered in ClinicalTrials.gov.

NCT Number	Country	Reference	Status	Intervention	Number of Participants	Age of Participants (Years)	Objective	Main Outcomes
NCT02256592	USA	–	Completed	300 mL of fecal suspension from a healthy donor delivered by colonoscopy and provided by OpenBiome	18	18–75	To examine the safety and durability of the intervention To determine the effects in ART-treated HIV-infected patients on: Immune activation Inflammatory biomarkers.	
NCT03329560	USA	–	Active, not recruiting	PRIM-DJ2727 (capsules containing lyophilized microbiota derived from 150 g of healthy donor stool) 1 daily dose 6 weeks	6(enrolled)	≥18	To evaluate whether the intervention is safe for ART-treated HIV-infected MSM.	
NCT03008941	Spain	Serrano-Villar et al., 2021 [80]	Completed	FT capsules provided by OpenBiome10 capsules first week + 5 capsules weekly8 weeks	30	18–80	To assess the effects in ART-treated HIV-infected patients on: bacterial metabolism plasma metabolite fingerprint immune function and activation inflammatory markers markers of enterocyte barrier function.	Safe. Modification of microbiota composition. No effects on CD4^+^ T lymphocytes, CD4/CD8 ratio, lymphocyte markers of immune activation, or plasma markers of inflammation and BT. Decreased IFABP levels.
NCT04165200	Mexico	–	Completed	FT via frozen capsules15 capsules every 12 h for four doses 7 days prior ART start and on weeks 0, 4, 8 and 12 after ART start	22	≥18	To monitor the effects in naive HIV-infected patients on: CD4^+^ T lymphocytes viral load during weeks 0, 4, 8, 12, and 24 after initiating ART.	

ART, antiretroviral treatment; BT, bacterial translocation; FT, fecal transplantation; HIV, human immunodeficiency virus; IFABP, intestinal fatty acid-binding protein; MSM, men who have sex with men.

**Table 6 ijerph-18-06837-t006:** Summary of the interventions described in this review and their composition.

	Intervention	Definition/Composition
**Prebiotics**	scGOS	Short-chain galacto-oligosaccharides
	lcFOS	Long-chain fructo-oligosaccharides
	pAOS	Pectin hydrolysate-derived acidic oligosaccharides
	Glutamine	Glutamine aminoacidic
	Camu camu	*Myrciaria dubia*
**Probiotics**	Supplemented fermented skimmed milk	*L. rhamnosus* GG (10^8^ CFU/mL), *B. animalis* subsp. *lactis* B-12 (10^8^ CFU/mL), and *L. acidophilus* La-5 (10^7^ CFU/mL)
	Visbiome	*S. thermophilus*, *B. breve*, *B. longum*, *B. infantis*, *L. plantarum*, *L. acidophilus*, *L. paracasei*, and *L. bulgaricus* (9 × 10^11^ CFU)
	*Saccharomyces boulardii*	56.5 mg of living *S. boulardii*
	*Lactobacillus rhamnosus HN001* plus *Bifidobacterium lactis Bi-07*	*L. rhamnosus HN001* plus *B. lactis Bi-07* (10^9^ CFU/mL)
	*Lactobacillus rhamnosus* GG	*L. rhamnosus* GG (10^10^ CFU)
	Packet A	*S. salivarius* spp. *thermophilus* (2.04 × 10^14^ CFU), *B. breve*, *B. infantis*, *B. longum* (9.3 × 10^13^ CFU), *L. acidophilus* (2 × 10^12^ CFU), *L. plantarum* (2.2 × 10^8^ CFU), *L. casei* (2.2 × 10^8^ CFU), *L. delbrueckii* ssp. *bulgaricus* (3 × 10^8^ CFU), and *S. faecium* (3 × 10^7^ CFU)
	*Lactobacillus rhamnosus* GR-1 and *Lactobacillus reuteri* RC-14	*L. rhamnosus* GR-1 and L. *reuteri* RC-14
	Ganeden BC^30^	*B. coagulans* GBI-30, 6086 (2 × 10^12^ CFU)
	Fortified yogurt	*L. rhamnosus* GR-1 (1.23 × 10^9^ CFU/mL)
	Biola	*L. rhamnosous* GG 10^8^ CFU, *L. acidophilus* La-5 10^7^ CFU and *B. animalis* subsp. *lactis* Bb-12 10^8^ CFU
	VSL#3	*S. thermophilus*, *B. breve*, *B. longum*, *B. infantis*, *L. plantarum*, *L. acidophilus*, *L. paracasei* and *L. bulgaricus* (9 × 10^11^ CFU)
	*Lactobacillus rhamnosus* GG plus *Lactobacillus acidophilus plus Lactobacillus bulgaricus plus Bifidobacterium* *animalis* subsp. *lactis* plus *Streptococcus thermophilus*	*L. rhamnosus* GG, *L. acidophilus*, *L. bulgaricus*, *B. animalis* subsp. *lactis*, and *S. thermophilus*
	*Vivomixx*	*B. breve*, *B. longum*, *B. infantis*, *L. acidophilus*, *L. plantarum*, *L. paracasei*, *L. bulgaricus*, and *S. thermophilus*
**Symbiotics**	Agavins plus *Lactobacillus rhamnosus* HN001 plus *Bifidobacterium lactis* Bi-07	10 g of agavins from *Agave tequilana* plus *Lactobacillus rhamnosus* HN001 and *Bifidobacterium lactis* Bi-07 (10^9^ CFU/mL)
	Synbiotic 2000	4 probiotic bacteria (10^10^ CFU/bacteria) plus 4 dietary fibers (2.5 g/fiber)
	PMT25431	Unknown
	LACTOFOS	*L. paracasei* Lpc-37 SD 5275, *Lactobacillus rhamnosus* HN001 SD 5675*, L. acidophilus* N CFM SD 5221, and *B. bifidum* (10^9^ CFU/strain) plus FOS
	Rillus	*L. plantarum* (8.55 mg), *S. thermophilus* (8.55 mg), and *B. bifidum* 2.55 mg (10^9^ CFU) plus FOS (480 mg)
	I3.1	*L. plantarum CECT7484, L. plantarum CECT7485* and *P. acidilactici)* plus ProSheed
	*Lactobacillus paracasei* plus *Lactobacillus rhamnosus* plus *Lactobacillus acidophilus* plus *Bifidobacterium lactis* plus FOS	*L. paracasei*, *Lactobacillus rhamnosus*, *L. acidophilus*, and *B. lactis* plus FOS

## Data Availability

Non applicable.

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
