# Peer review of "Lights and Shadows of Microbiota Modulation and Cardiovascular Risk in HIV Patients"

_ijerph, 2021, doi:10.3390/ijerph18136837_

Round 1

Reviewer 1 Report

In the last 10 years, the topic on intestinal microbiota/microbioma disturbance and it’s impact on the risk of several comorbidities as it’s CVD, diabetes, obesity in HIV patients (also in the general population) is emerging as one of the most important matter on the management of this chronic diseases. In this scenario we need to have better knowledge on the physiopathology and interventions on modulating GM composition to counteract the consequences of dysbiosis and the outcomes of systemic inflammation and the related accelerated aging.

In my knowledge this is the first review to investigate if it exists a solid scientific evidence that supports the use of prebiotics, probiotics, symbiotic and FT to modify GM in HIV-infected patients.

The conclusions are very useful giving us clues how to proceed the research to know more about the use of these biotics along with the standard of care to modify CVR in HIV-patients.

I would like to add that the authors of this manuscript have a solid research in gut microbiota composition on HIV-infected patients that contributed to provide a clear statement of the problem, interpreting the findings, and stating the conclusions.

Finally I have minor comments:

In order to be better understandable to the readers it will be useful if the authors present two tables summarizing one the definition of the biotics and the other with the list of bacteria of the health intestinal microbiota and those associated to dysbiosis, bacterial translocation, inflammation and CVR in HIV-patients.

Author Response

In the last 10 years, the topic on intestinal microbiota/microbioma disturbance and its impact on the risk of several comorbidities as it is CVD, diabetes, obesity in HIV patients (also in the general population) is emerging as one of the most important matter on the management of this chronic diseases. In this scenario, we need to have a better knowledge on the physiopathology and interventions on modulating GM composition to counteract the consequences of dysbiosis and the outcomes of systemic inflammation and the related accelerated aging. In my knowledge, this is the first review to investigate if it exists a solid scientific evidence that supports the use of prebiotics, probiotics, symbiotic and FT to modify GM in HIV-infected patients. The conclusions are very useful giving us clues how to proceed the research to know more about the use of these biotics along with the standard of care to modify CVR in HIV-patients. I would like to add that the authors of this manuscript have a solid research in gut microbiota composition on HIV-infected patients that contributed to provide a clear statement of the problem, interpreting the findings, and stating the conclusions.

Response: We appreciate very much the reviewer positive comments about our manuscript and our background. We are very pleased with your view. We strongly believe that it is essential to provide a solid scientific evidence of the potential usefulness of prebiotics, probiotics, symbiotics and fecal transplantation -along with the standard of care- to reduce the higher cardiovascular risk seen in HIV population (with dysbiosis).

In order to be better understandable to the readers it will be useful if the authors present two tables summarizing one: the definition of the biotics and the other: with the list of bacteria of the health intestinal microbiota and those associated to dysbiosis, bacterial translocation, inflammation and CVR in HIV-patients.

Response: As suggested, we have added a table summarizing the different biotics included in this review with their definition or composition (Table 6, page 19) and a second table summarizing the bacteria whose relative abundance has been modified by the interventions described in this review. We have also included a brief description of their functions in health and disease (Table 4, page 23). We agree with the reviewer that these tables will clearly improve our manuscript.

Reviewer 2 Report

This review is interesting and represents a field of study that is relevant, but I feel as if a review on this specific topic is premature in nature. There are only two trials reviewed for the “prebiotics” section, and while there are a great number of different kinds and origins of prebiotics, only a few are studied here and seem to have been used in a combo-type capacity, so attributions of specific effects of specific prebiotics cannot be discerned. Additionally, in general, I don’t believe the authors do a good job of explaining or postulating the connection between the gut microbiota and cardiovascular risk in HIV patients in general. This review may have more potential for publication as an analysis of how these treatments affect the gut microbiota and immune functioning in HIV patients rather than focusing on cardiovascular risk.

Author Response

This review is interesting and represents a field of study that is relevant, but I feel as if a review on this specific topic is premature in nature. There are only two trials reviewed for the “prebiotics” section, and while there are a great number of different kinds and origins of prebiotics, only a few are studied here and seem to have been used in a combo-type capacity, so attributions of specific effects of specific prebiotics cannot be discerned. Additionally, in general, I do not believe the authors do a good job of explaining or postulating the connection between the gut microbiota and cardiovascular risk in HIV patients in general. This review may have more potential for publication as an analysis of how these treatments affect the gut microbiota and immune functioning in HIV patients rather than focusing on cardiovascular risk.

Response: We are very sorry to hear these comments. Our review has clearly revealed that there is no solid scientific evidence to recommend the use of prebiotics, symbiotics and even probiotics in HIV-infected patients in order to reduce the incidence of co-morbidities such as cardiovascular events. In fact, the small number of studies published in this field (for example in the “prebiotics” section) cannot be seen as a limitation of our review but the proof that more scientific evidence is needed before generalizing. In addition, the huge variability observed in the type of compounds, the dose and the length of administration observed among the trials also raises serious questions about the practical value of these approaches. These are our conclusions and should serve to design new clinical trials focused on this topic with clear implications on HIV-infected patients´ management.

Concerning the connection between the gut microbiota and cardiovascular risk in HIV patients, we agree with the reviewer that we have presented it in a very general way in the first section of the manuscript. We believe that the focus of our review was to provide evidence of the effects of different biotics on HIV-infected people and their effects on cardiovascular risk more than to explain the molecular mechanisms underlying such association. However, if the reviewer still believes it is necessary to include a specific section specifically focus on this topic, we would be pleased to do it.

Finally, and as stated to the Editor, we also believe that a review focused on “microbiota and immune function in HIV patients” could be of interest, however, there are several reviews already published in this field. In fact, a low diversity of microbiota combined with the outgrowth of pathogenic bacterial species together with dysregulated metabolic pathways have been linked to compromised gut immunity, bacterial translocation and systemic inflammation (El-Far & Tremblay, Curr Opin HIV AIDS, 2018), which translate into increased risk of cardiovascular events. Specifically, the increased cardiovascular risk could have a significant clinical relevance as well as a significant impact on patient management. Thus, our purpose was to review this new topic with a clear clinical relevance.

Extensive editing of English language and style required

Response: Following the reviewer suggestion, a native English speaker has reviewed the new version of the manuscript in order to improve the language.

Reviewer 3 Report

In this review article, the authors reviewed and discussed the results obtained from 3 clinical trials focused on prebiotics, 25 concerning probiotics, 6 concerning symbiotics and 4 concerning fecal transplantations. The authors found that none of these trials investigated if these compounds were able to reduce CVD in HIV patients.

Comments

This is an interesting review article. This manuscript is well-written. The reviewer has only some minor concerns as follows:

  1. In Figure 1, the words in the purple boxes are too small to viewable. Moreover, the indications for cell types are not easy to distinguish.
  2. In Tables 1-3, in the “associated article” fields, it can be changed to: like as “Serrano-Villar et al., 2017 (24)”.

Author Response

In this review article, the authors reviewed and discussed the results obtained from 3 clinical trials focused on prebiotics, 25 concerning probiotics, 6 concerning symbiotics and 4 concerning fecal transplantations. The authors found that none of these trials investigated if these compounds were able to reduce CVD in HIV patients. This is an interesting review article. This manuscript is well-written.

Response: We appreciate the reviewer´s positive comments. We hope that the changes suggested and carried out in the present version of the manuscript will improve it.

In Figure 1, the words in the purple boxes are too small to viewable. Moreover, the indications for cell types are not easy to distinguish.

Response: We have changed Fig1 following the reviewer suggestion and we have now presented it horizontally to obtain bigger words (page 3). Besides, we have increased its resolution. We think that now it is much more viewable and that all cell types are distinguishable.

In Tables 1-3, in the “associated article” fields, it can be changed to: like as “Serrano-Villar et al., 2017 (24)”

Response: As suggested, we have changed the column “associated article” following the indications from the reviewer.